# The Inhibitory Effects of Gold Nanoparticles on VEGF-A-Induced Cell Migration in Choroid-Retina Endothelial Cells

**DOI:** 10.3390/ijms21010109

**Published:** 2019-12-23

**Authors:** Chi-Ming Chan, Chien-Yu Hsiao, Hsin-Ju Li, Jia-You Fang, Der-Chen Chang, Chi-Feng Hung

**Affiliations:** 1School of Medicine, Fu-Jen Catholic University, New Taipei City 24205, Taiwan; chancm@mail.fju.edu.tw (C.-M.C.); sakumanatsumi@gmail.com (H.-J.L.); 2Department of Ophthalmology, Cardinal Tien Hospital, New Taipei City 23148, Taiwan; 3Department of Nutrition and Health Sciences, Research Center for Food and Cosmetic Safety, and Research Center for Chinese Herbal Medicine, College of Human Ecology, Chang Gung University of Science and Technology, Taoyuan 33303, Taiwan; mozart@gw.cgust.edu.tw; 4Aesthetic Medical Center, Department of Dermatology, Chang Gung Memorial Hospital, Taoyuan 33305, Taiwan; 5Pharmaceutics Laboratory, Graduate Institute of Natural Products, Chang Gung University, Taoyuan 33303, Taiwan; fajy@mail.cgu.edu.tw; 6Department of Mathematics and Statistics and Department of Computer Science, Georgetown University, Washington, DC 20057, USA; Chang@georgetown.edu; 7Ph.D. Program in Pharmaceutical Biotechnology, Fu-Jen Catholic University, New Taipei City 24205, Taiwan; 8MS Program in Transdisciplinary Long Term Care, Fu-Jen Catholic University, New Taipei City 24205, Taiwan

**Keywords:** gold nanoparticles (AuNPs), vascular endothelial growth factor (VEGF), cell migration, Akt, endothelial nitric oxide synthase (eNOS), choroidal and retinal neovascularization

## Abstract

Background: Vascular endothelial growth factor (VEGF) is upregulated by hypoxia and is a crucial stimulator for choroidal neovascularization (CNV) in age-related macular degeneration and pathologic myopia, as well as retinal neovascularization in proliferative diabetic retinopathy. Retinal and choroidal endothelial cells play key roles in the development of retinal and CNV, and subsequent fibrosis. At present, the effects of gold nanoparticles (AuNPs) on the VEGF-induced choroid-retina endothelial (RF/6A) cells are still unknown. In our study, we investigated the effects of AuNPs on RF/6A cell viabilities and cell adhesion to fibronectin, a major ECM protein of fibrovascular membrane. Furthermore, the inhibitory effects of AuNPs on RF/6A cell migration induced by VEGF and its signaling were studied. Methods: The cell viability assay was used to determine the viability of cells treated with AuNPs. The migration of RF/6A cells was assessed by the Transwell migration assay. The cell adhesion to fibronectin was examined by an adhesion assay. The VEGF-induced signaling pathways were determined by western blotting. Results: The 3-(4,5-Dimethylthiazol-2-yl)-2,5-diphenyltetrazolium bromide (MTT) viability assay revealed no cytotoxicity of AuNPs on RF/6A cells. AuNPs inhibited VEGF-induced RF/6A cell migration in a concentration-dependent manner but showed no significant effects on RF/6A cell adhesion to fibronectin. Inhibitory effects of AuNPs on VEGF-induced Akt/eNOS were found. Conclusions: These results suggest that AuNPs are an effective inhibitor of VEGF-induced RF/6A cell migration through the Akt/eNOS pathways, but they have no effects on their cell viabilities and cell adhesion to fibronectin.

## 1. Introduction

Angiogenesis is the physiological process involving the growth of new blood vessels from existing vasculature [1]. It plays a central role in cancer and various ischemic diseases [2,3]. Vascular endothelial growth factor (VEGF) is upregulated by hypoxia during ischemia [4] and is a major stimulatory factor for choroidal neovascularization (CNV) in age-related macular degeneration [5,6] and high myopia [7], as well as retinal neovascularization in diabetic retinopathy [6]. Retinal and choroidal endothelial cells play key roles in the development of retinal and choroidal neovascularization, and subsequent fibrosis.

The migration of endothelial cells (EC) is a notable and key step in angiogenesis [8], but the detailed signaling molecules responsible for this migration are still under investigation. Although numerous factors affect EC migration, the influence of VEGF has received considerable attention from researchers, as VEGF is widely used as a signaling molecule for the induction of EC migration [9,10].

Gold nanoparticles (AuNPs) have been shown to be the material of choice of many diagnostic platforms. Their unique electronic, biocompatible, and molecular-recognition properties of small-sized AuNPs broaden the potential of gold in several fields of application [11]. Gold nanoparticles displayed an anti-angiogenic effect. They induce nanostructural reorganization of VEGFR2 on the human umbilical vascular endothelial cells (HUVEC) to repress angiogenesis [12]. Moreover, AuNPs can significantly inhibit HepG2-conditioned medium (HepG2-CM) activated HUVEC proliferation and migration through the down-regulation of VEGF activity and disruption of cell morphology [13]. AuNPs were shown to inhibit VEGF-induced migration in HUVEC [14] and laser-induced CNV in mice [15]. Fabrication of resveratrol-coated gold nanoparticles can increase the retinal pigment epithelium-derived factor and decrease the VEGF-1 in streptozotocin-induced diabetic rats [16]. However, the effects of AuNPs on the choroid-retina endothelial (RF/6A) cells viability and VEGF-induced RF/6A cell migration are still unknown.

In the present study, we investigated the inhibitory effect of AuNPs on VEGF-induced RF/6A cell migration and the possible underlying mechanisms involved. These mechanisms include the influence of AuNPs on RF/6A cell viability, cell adhesion, and Akt/endothelial nitric oxide synthase (eNOS) pathway activation.

## 2. Results 

### 2.1. AuNPs Showed No Cytotoxicity on RF/6A Cells

To eliminate the possibility that AuNPs have an effect on RF/6A cell migration through their effects on cell viability, cell viability was determined by MTT assays. As shown in Figure 1, the treatment of AuNPs (1, 2, and 4 ppm) did not change cell viability in MTT assays. These data show that AuNPs have no cytotoxicity toRF/6A cells and their effects on cell migration did not result from the reduction of cell viability.

### 2.2. AuNPs Suppressed VEGF-Induced RF/6A Cell Migration

To decide the inhibitory activities of AuNPs on RF/6A cell migration, we carried out the Transwell migration assays. The data indicate that cell migration of RF/6A was increased by VEGF, and this effect was prominently inhibited by the preincubation of VEGF with AuNPs in a concentration-dependent manner. However, AuNPs had no effect on basal RF/6A cell migration without the treatment of VEGF (Figure 2). 

### 2.3. AuNPs Had No Effect on RF/6A Cell Adhesion

To decide whether AuNPs suppressed RF/6A cell migration by interfering with their attachment to fibronectin, we conducted the effect of AuNPs on RF/6A cell adhesion with fibronectin-coated. As shown in Figure 3, the adhesion number was not affected by the treatment of AuNPs. These data suggest that the suppression of AuNPs on RF/6A cell migration was not produced by interference with the attachment of the cells to fibronectin.

### 2.4. AuNPs Suppressed VEGF-Induced Akt and eNOS Phosphorylation

To decide whether VEGF-induced signaling pathways are influenced by AuNPs, the level of phosphorylation of Akt and eNOS was determined by Western blotting. Figure 4 indicates that Akt and eNOS phosphorylation were enhanced by the treatment of VEGF. Preincubation with AuNPs produced the decrease of VEGF-induced PI3K and Akt phosphorylation in a dose-dependent manner.

## 3. Discussion 

Angiogenesis is the physiological process of forming new blood vessels from preexisting vasculature. Physiological angiogenesis is highly regulated during wound repair [17,18]. Pathological choroidal and retinal angiogenesis are the leading causes of blindness, including age-related macular degeneration and diabetic retinopathy [19,20]. In angiogenesis, a serial process participated with several cells that include proteolytic degradation of the extracellular matrix, followed by migration and proliferation of capillary endothelial cells, pericyte recruitment, and assembly of the mature vessel [21]. The angiogenic process is regulated by a tight balance between pro-, and anti-angiogenic agents and vascular endothelial growth factor (VEGF) plays a critical regulatory role [22]. It is physiologically required for regulating proliferation and assembling endothelial cells during vasculogenesis, as well as for their maintenance and survival throughout the lifetime of blood vessels [23]. However, under subtle pathological alterations, abnormal angiogenesis with retinal and choroidal microvascular alterations have been observed during PDR, high myopia, and neovascular AMD [24,25,26].

AuNPs have been explored with a high expectation as they are effective and promising agents to improve the diagnosis and treatment of cancer [27,28,29]. Gold nanoparticles downregulate cellular cascades of interleukin-1β-induced pro-inflammatory response [30], and topical application of AuNPs decreases intraocular oxidative damage and inflammation [31]. Besides, AuNPs can effectively inhibit matrix metalloproteinase activity without causing cytotoxicity or inflammation [32]. AuNPs induce oxidative stress in mouse fibroblast, and the cells trigger the autophagic pathways as a survival mechanism to avoid cell death [33]. Gold nanoparticles can also inhibit retinal neovascularization through autophagy [34]. AuNPs can interrupt the crosstalk of tumor microenvironment and endothelial cells through the blockade of VEGF-VEGFR2 signaling during angiogenesis [35]. Under flow exposure conditions, anti-intercellular adhesion molecule-1 (ICAM-1) AuNPs can activate leukocyte adhesion receptors in tumor necrosis factor (TNF)-activated shear stress-adapted endothelial cells [36].

Many studies showed that AuNPs do not affect cell viabilities in several cell types [37], but some studies demonstrated their toxic effects [38,39]. The effect of AuNPs on choroid-retina endothelial cell viability is still unknown. Our study demonstrated that there are no toxic effects on RF/6A cell viabilities by AuNPs. Endothelial cell migration is essential to angiogenesis [40]. This motile and directional process is controlled by chemotactic, the directional migration toward a gradient of soluble chemoattractant; haptotaxis, the directional migration toward a gradient of immobilized ligands; and mechanotaxis, the directional migration generated by mechanical forces [41]. During angiogenesis, endothelial cell migration is the integrated as a result of the above three mechanisms. In this study, we found that AuNPs significantly inhibit VEGF-induced choroid-retinal endothelial RF/6A cell migration without any signs of cytotoxicity as well.

Fibronectin is the main structural component of internal/outer collagenous and the elastic layer of Bruch’s membrane [42]. It is also one of the serum autoantibody biomarkers for neovascular age-related macular degeneration [43]. In epiretinal membranes of vitreoproliferative retinopathy and proliferative diabetic retinopathy under immunohistochemical study, fibronectin is a major component in the extracellular matrix [44]. Under the retinal ischemic conditions, fibronectin is upregulated [45]. It is required for endothelial cell migration and tube morphogenesis and plays an important role during retinal and choroidal neovascularization [46]. However, our results indicate that AuNPs does not affect choroid-retinal endothelial RF/6A cell adhesion to fibronectin. 

The PI3K/Akt pathway provides essential signaling for cell survival and proliferation. Signaling from different eNOS agonists, such as VEGF, insulin, and estrogen can affect eNOS activity through the PI3K/AKT pathway [47,48,49]. It has been reported that activations of the survival signal PI3K/Akt pathway and the endothelial-specific eNOS/NO pathway were closely associated with vascular remodeling and angiogenesis [50,51]. However, whether AuNPs affects the biological properties of choroid-retinal endothelial cells and the role of/Akt/eNOS signaling pathway in *VEGF*-induced migration, have remained poorly understood. In the present study, we determined the effects of AuNPs on the *VEGF*/Akt/eNOS signaling pathway on choroid-retinal endothelial RF/6A cell migration. The results showed that AuNPs suppressed VEGF-induced activation of the Akt/eNOS signaling pathway during these processes. 

## 4. Material and Methods 

### 4.1. Materials

Gold nanoparticles (AuNPs) were purchased from Gold NanoTech Inc. (Taipei, Taiwan) [52]. The synthesis methods of AuNPs have been described in our previous study [53]. The size of the AuNPs we used in this study was 3–5 nm (SA20130129b). Phenylmethylsulfonyl fluoride (PMSF), bovine serum albumin (BSA), leupeptin, aprotinin, sodium fluoride (NaF), and sodium orthovanadate were purchased from Sigma-Aldrich (St Louis, MO, USA). Antibodies (Ab) raised against phospho-eNOS and eNOS were from Santa Cruz Biotechnology (Santa Cruz, CA, USA). Abs raised against phospho-Akt and Akt were from Cell Signaling Technology, Inc. (Beverly, MA, USA). 

### 4.2. Cell Cultures 

The rhesus macaque choroid-retinal endothelial cell line RF/6A derived from the choroid-retina of a rhesus macaque fetus was purchased from Food Industry Research and Development Institute (Hsinchu, Taiwan). The cells were maintained in RPMI 1640 Medium (Gibco, Carlsbad, CA, USA) supplemented with 10% fetal bovine serum (FBS, Gibco) and 1% penicillin/streptomycin (Hyclone, UT, USA) at 37 °C, 5% CO_2_ and 95% humidified air. For most of the experiments, cells reaching a 90–95% of confluence were synchronized for 24 h by serum starvation before they were subjected to further analysis. 

### 4.3. Nanogold Treatment and VEGF Incorporation 

AuNPs were dissolved by distilled water in a series of dilutions. In Transmigration assays and the Western blot analysis, the 25 ng/mL of vascular endothelial growth factor (VEGF) were incorporated with different concentrations of AuNPs in the serum-free RPMI 1640 medium in culturing RF/6A cells at 37 °C for 30 min.

### 4.4. Transmigration Assays 

By using a modified Boyden chamber model (Transwell apparatus, 8.0 mm pore size, Costar), the migration ability of RF/6A cell were demonstrated as previously described [54]. Briefly, the lower surface of the semi-permeable membrane was coated with 0.3 mg of fibronectin for 30 min in the laminar flow hood. Different concentrations of AuNPs were added to the lower chamber, which was filled with 0.6 mL of serum-free RPMI 1640 medium or 25 ng/mL VEGF-containing medium. 2.5 × 10^5^/mL RF/6A cells were added to the upper chamber. Inserts were removed after 5 h, and the inner side was wiped with cotton swabs. Cells at the lower chamber were fixed and stained with 0.5% toluidine blue in 4% paraformaldehyde (PFA). The migrated cells were photographed and counted as the number of stained cells per ×100 field (high power field, HPF) under a phase-contrast microscope (Leica DMIL1) and photographed.

### 4.5. Viability Assays

The cell viability was detected by the 3-(4,5-dimethylthiazol-2-yl)-2,5-diphenyltetrazolium bromide (MTT) assay. Briefly, different concentrations of AuNPs were preincubated with RF/6A cells for 24 h in 96-well plates. After a brief wash with medium, 0.5 mg/mL MTT was added and incubated at 37 °C for 4 h to measure the amount of living and metabolically active cells. MTT was metabolized by mitochondrial dehydrogenases to a purple formazan dye, with the light absorbance at 570 nm. The absorbance was detected by a spectrophotometer and was proportional to cell viability.

### 4.6. Cell Adhesion Assays

The 50 µL of fibronectin (15 µg/mL in PBS, pH 7.4) was used to coat 96-well plates at 37 °C for 24 h. After gentle washed with PBS three times, the plates were blocked by 100 mg/mL bovine serum albumin (Sigma-Aldrich, St Louis, MO, USA) in PBS at room temperature for 1 h to avoid nonspecific binding to fibronectin. Afterward, RF/6A cells were labeled with 10 mg/mL BCECF/AM for 30 min at 37 °C in serum-free RPMI 1640 medium. After brief washed with the serum-free medium twice, the labeled cells were resuspended to a density of 1.0 × 10^5^ and incubated with different concentrations of AuNPs in the serum-free RPMI 1640 medium for further 30 min at 37 °C. Then, the suspended cells were plated onto 96-well plates with 100 µL serum-free cell culture medium at 37 °C for 1 h. The nonadherent cells were removed from the plate by washing with PBS for three times and aspirated. The number of adhered cells in the 96-well plates was measured by the Wallac Victor 3 1420 multilabel counter (Perkin Elmer, Turku, Finland) at an excitation and emission wavelength of 485 and 535 nm. 

### 4.7. Preparation of Cell Lysates for Western Blots 

The RF/6A cells were cultured on 6 cm dishes until 90–95% confluent and changed into a serum-free RPMI 1640 medium for 24 h. Various concentrations of nanogold mixed with or without VEGF (25 ng/mL) at 37 °C for 30 min in serum-free medium were added to the cells. After 30 min of incubation, cell were washing with PBS twice and lysing with sonication and centrifugation at 14,000 g for 10 min at 4 °C in the radioimmunoprecipitation assay buffer. The supernatant was removed and the concentration of protein was quantified by a Pierce protein assay kit (Pierce, Rockford, IL, USA). Total protein was separated by electrophoresis on 10% SDS–polyacrylamide gels. The protein bands were then transferred onto polyvinylidene fluoride (PVDF) membrane and probed with the antibodies against the phosphorylation of Akt and eNOS. Signals were detected by enhanced chemiluminescence (Chemiluminescence Reagent Plus from NEN, Boston, MA, USA). The PVDF membrane was stripped at 60 °C for 30 min with a stripping buffer for staining internal control.

### 4.8. Statistical Analysis

All results were analyzed with SigmaPlot for Windows (Version 10.00, Chicago, IL, USA). Data are shown as mean ± standard error (SE) of four experiments. The *t*-test was performed to determine the difference between groups. A value of *p* < 0.05 was considered statistically significant.

## 5. Conclusions 

As VEGF is a key mediator and upregulated in many ocular angiogenic diseases such as proliferative diabetic retinopathy and wet-type age-related macular degeneration, anti-VEGF therapy is the current main strategy of treatment. Our studies provide the mechanism of AuNPs on the inhibitory effects of the VEGF-induced RF/6A cell migration through the suppression of Akt/eNOS phosphorylation. However, AuNPs show no cytotoxic effects on the RF/6A cells. Moreover, AuNPs do not affect the normal physiological functions of RF/6A cell adhesion to fibronectin. These studies provide important insight into further research on the effects of AuNPs in RF/6A cells and the possible beneficial effect on the suppression of angiogenesis by AuNPs in ocular angiogenic diseases.

## Figures and Tables

**Figure 1 ijms-21-00109-f001:**
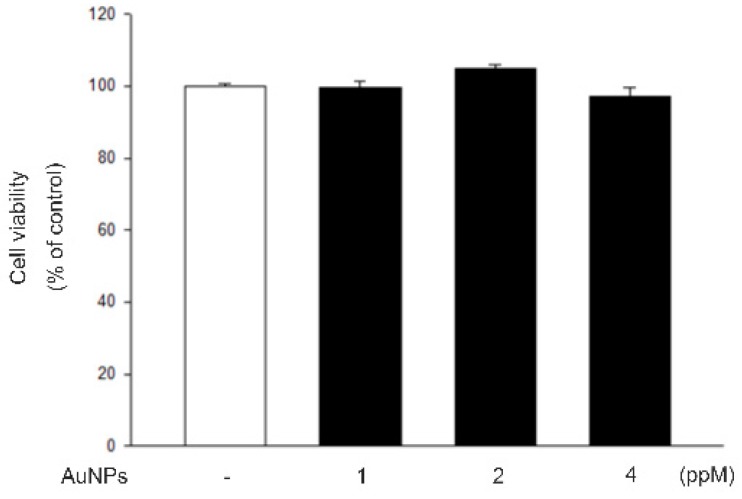
Viability of RF/6A cells was not influenced by AuNPs. The cells were treated with different concentrations of AuNPs for 24 h after being starved for 24 h. Cell viability was determined by the 3-(4,5-Dimethylthiazol-2-yl)-2,5-diphenyltetrazolium bromide (MTT) assay. The results are expressed as a percentage of control and represent the mean ± standard errors (SE) of four independent experiments.

**Figure 2 ijms-21-00109-f002:**
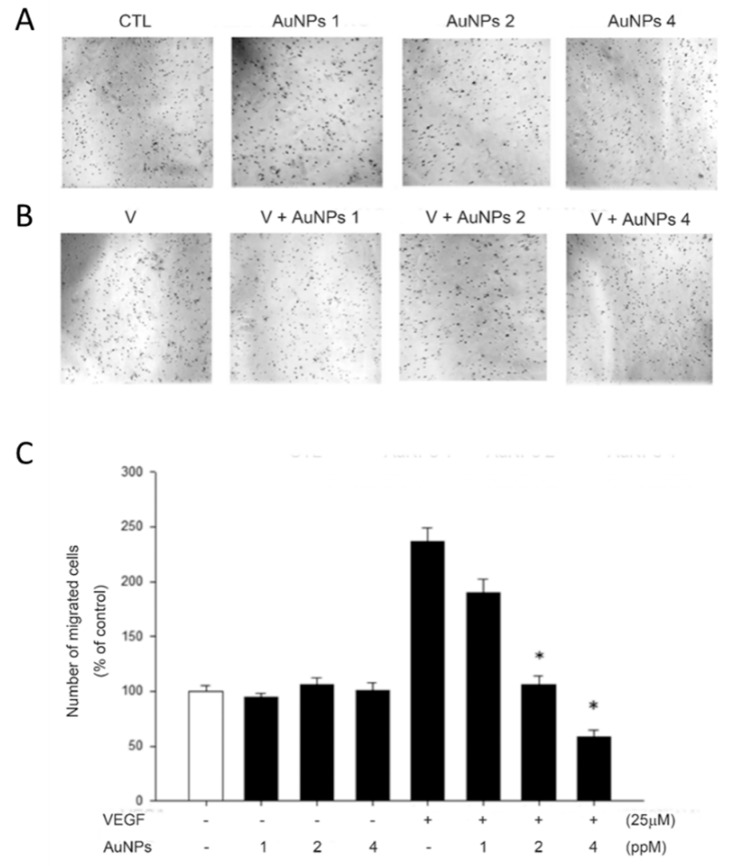
AuNPs suppress VEGF-induced cell migration in RF/6A cells by the Transwell migration assay. The Transwell inserts were coated with fibronectin (0.3 mg). RF/6A cells (5 × 10^4^ in 200 μL) were seeded in the upper chamber in the absence or presence of AuNPs. The inserts were assembled in the lower chamber, which was filled with 600 μL serum-free medium without VEGF, (**A**) containing VEGF (25 ng/mL), (**B**) and preincubated with various concentrations of AuNPs for 30 min at 37 °C. After incubating for 5 h at 37 °C, fixation was performed. RF/6A cells that migrated to the underside of the filter membrane were photographed (A, B) and counted by phase-contrast light microscope under high power field (magnification, 100×); (**C**) All experiments were conducted in duplicates, and similar results were repeated four times. The results are expressed as a percentage of control and represent the mean ± standard errors (SE) of the eight experiments. * *p* < 0.05 significantly differs from VEGF-stimulated cells (the fifth bar).

**Figure 3 ijms-21-00109-f003:**
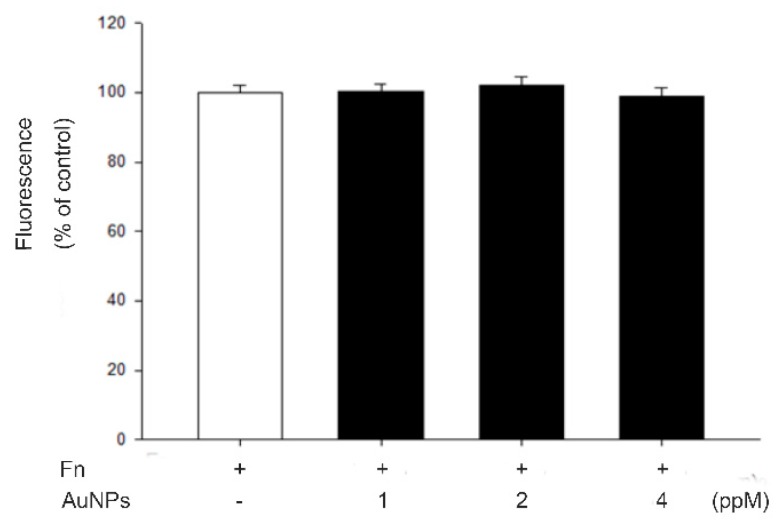
Cell adhesion of RF/6A cells was not influenced by AuNPs. BCECF-labeled cells were treated with DMSO or nanogold for 30 min. They were then seeded and allowed to adhere to plates with precoated fibronectin (fn) (15 µg/mL) at 37 °C for 1 h. Fluorescence was measured using excitation and an emission wavelength of 485 and 535 nm, respectively. The results are expressed as a percentage of control and represent the mean ± standard errors (SE) of three independent experiments.

**Figure 4 ijms-21-00109-f004:**
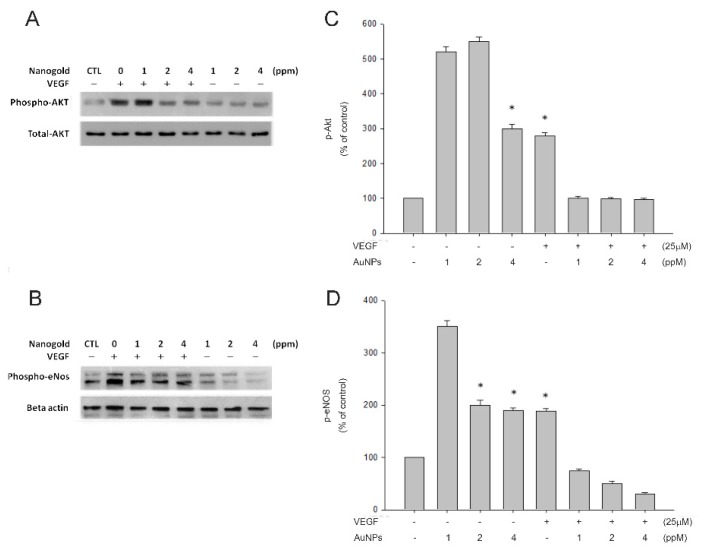
VEGF-induced protein kinase B (Akt) and endothelial nitric oxide synthase (eNOS) phosphorylations were inhibited by AuNPs. RF/6A cells were preincubated with the indicated concentrations of AuNPs (1, 2, 4 ppm) and incubated with or without VEGF (25 ng/mL) at 37 °C for 30 min, the cells were collected, and their lysates were analyzed by Western blot analysis. The changes in phosphorylated Akt and eNOS expression were evaluated (**A**,**C**). The quantitative data of western blot are shown below the panels, which are expressed as a percentage of control and represent the mean ± standard errors (SE) of the three independent experiments (**B**,**D**). * *p* < 0.05 significantly differs from VEGF-stimulated cells (the second bar) (**B**,**D**).

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
