# Peer review of "The Inhibitory Effects of Gold Nanoparticles on VEGF-A-Induced Cell Migration in Choroid-Retina Endothelial Cells"

_ijms, 2019, doi:10.3390/ijms21010109_

Round 1
Reviewer 1 Report
AuNPs do not affect the normal physiological functions of RF/6A cell adhesion to fibronectin. These results suggest that AuNPs are an effective inhibitor of VEGF-induced RF/6A cell migration through Akt/eNOS pathways, but they have no effects on the cell viabilities and cell adhesion to fibronectin. These studies may provide an important insight on further research on the effects of AuNPs in RF/6A cells and the possible beneficial effect on the suppression of angiogenesis by AuNPs in ocular angiogenic diseases.
Author Response
Reviewer 1
Comments and Suggestions for Authors:
AuNPs do not affect the normal physiological functions of RF/6A cell adhesion to fibronectin. These results suggest that AuNPs are an effective inhibitor of VEGF-induced RF/6A cell migration through Akt/eNOS pathways, but they have no effects on the cell viabilities and cell adhesion to fibronectin. These studies may provide an important insight on further research on the effects of AuNPs in RF/6A cells and the possible beneficial effect on the suppression of angiogenesis by AuNPs in ocular angiogenic diseases.
Response to Reviewer 1:
Thank you for your detailed review and constructive suggestions of our manuscript. We have rechecked the English language and style of this manuscript. Our corrections of this revision were highlighted blue. Thank you very much again.
Reviewer 2 Report
The authors should given more detail information about Gold Nanoparticles since different Synthesis methods or compositions of Gold Nanoparticles may have different characteristics which may be crucial for the cellular uptake and targeting etc.

Author Response
Reviewer 2
Comments and Suggestions for Authors:
The authors should given more detail information about Gold Nanoparticles since different Synthesis methods or compositions of Gold Nanoparticles may have different characteristics which may be crucial for the cellular uptake and targeting etc.
Response to Reviewer 2:
Thank you for your detailed review and constructive suggestions.
Gold nanoparticles (AuNPs, SA20130129b) were purchased from Gold NanoTech Inc. (Taipei, Taiwan). The synthesis methods of AuNPs have been described in our previous study[1]. The size of AuNPs we used in this study was 3–5 nm. AuNPs were made by physical manufacturing and contained no stabilizers or surface modifiers. Briefly, the gold was cut into the target material and vaporized to the atomic level by electrical gasification. Under high vacuum (10−8 Pa), the gold was evaporated and slowly deposited as AuNPs in distilled water. The sizes of the AuNPs were controlled by regulating the evaporation time and electric current intensity [2-4]. The maximal concentration of AuNPs of sizes 3–5nm were controlled at 100 ppm, without obvious aggregation.
We also have rechecked the English language and style of this manuscript. Our corrections of this revision were highlighted blue. Thank you very much again.
Lu, P. H.; Li, H. J.; Chang, H. H.; Wu, N. L.; Hung, C. F., Gold nanoparticles induce cell death and suppress migration of melanoma cells. J Nanopart Res 2017, 19, (10). Lai, T. H.; Shieh, J. M.; Tsou, C. J.; Wu, W. B., Gold nanoparticles induce heme oxygenase-1 expression through Nrf2 activation and Bach1 export in human vascular endothelial cells. Int J Nanomedicine 2015, 10, 5925-39. Tsai, Y. Y.; Huang, Y. H.; Chao, Y. L.; Hu, K. Y.; Chin, L. T.; Chou, S. H.; Hour, A. L.; Yao, Y. D.; Tu, C. S.; Liang, Y. J.; Tsai, C. Y.; Wu, H. Y.; Tan, S. W.; Chen, H. M., Identification of the nanogold particle-induced endoplasmic reticulum stress by omic techniques and systems biology analysis. ACS Nano 2011, 5, (12), 9354-69. Yen, H. J.; Hsu, S. H.; Tsai, C. L., Cytotoxicity and immunological response of gold and silver nanoparticles of different sizes. Small 2009, 5, (13), 1553-61.
